# WORDING IMAGE FOR DOMAIN-INVARIANT REPRESENTATION IN DOMAIN GENERALIZATION

## ABSTRACT

The out-of-domain (OOD) generalization ability is essential for AI systems in real-world applications. Recent works proposed to utilize prior knowledge like unlabeled data or descriptions to improve the OOD performance. However, the assumption that the domain of each test data is known in advance is unrealistic in practice, which limits the generalization of AI system to various domains and prevents the wide deployment. In this paper, we introduce WIDIn, which Words the Images to learn Domain-Invariant features being robust to complex domain biases. Different from visual embeddings, the language embeddings of class descriptions are domain-invariant, and they can be connected via vision-language alignment. Thus, we propose to project images into language space by representing each image as a word token, which is attached with hand-crafted prompt and fed into language encoder. Then, the difference between the extracted embedding and the language embedding of its class description is used to estimate the domain-specific counterpart, which facilitates the domain-invariant representation learning. Notably, our WIDIn can be applied to both pretrained vision-language models like CLIP, and separately trained uni-modal models like MoCo and BERT. Experimental studies on two domain generalization benchmark datasets and two long-tail benchmark datasets demonstrate the effectiveness of our approach.

## 1 INTRODUCTION

Recent years have witnessed the great success of large-scale pre-trained models for the application in various downstream tasks (He et al., 2016; 2020; Radford et al., 2021). Nevertheless, when adapting the pre-trained models via learning upon the train set of a new domain, the performance can meet the expectations still only when the test data also comes from the source domain seen in training. To mitigate the performance drop on the out-of-distribution data (Wang & Deng, 2018), domain adaptation (DA) has been investigated to transfer the representation learnt from source domain to the new target domains. However, traditional DA methods require the prior knowledge of target domain like unlabeled data, which could be unrealistic in data-sparse situation such as rare disease.

To reduce the dependence on knowledge from target domains, one-shot DA (Yang & Lim, 2020) is proposed to request additional one example per target domain to update the model, while language-guided DA (Dunlap et al., 2023) uses domain descriptions. In contrast, there is lack of studies in the scenario where the adaptation without any prior, *i.e.*, domain generalization (Zhou et al., 2021), in particular, when the train set is only from a *single* source domain. After all, the language-guided DA still needs the description before training while it is implausible to exhaust all possibility with precise depiction, *e.g.*, "quickdraw" in DomainNet (Peng et al., 2019). Furthermore, as the data can be of diverse domains at test time, annotating each test data with domain label is expensive.

To overcome the limitation caused by the requirement of knowledge in target domains, a desired system should be 1) robust to unseen domains and 2) independent from any prior & assumption of the test data. However, as the train set is from a single domain, the learned models may inevitably show strong domain bias which is invariant to classification (*e.g.*, airplane is almost gray in natural images but dogs can also be gray in drawing) and may also contain instance-specific information such as background. Thus, to improve the generalization, it is essential to unravel the *domain-invariant* representation being robust across domains (*e.g.*, the shape is essential for recognizing airplane) and

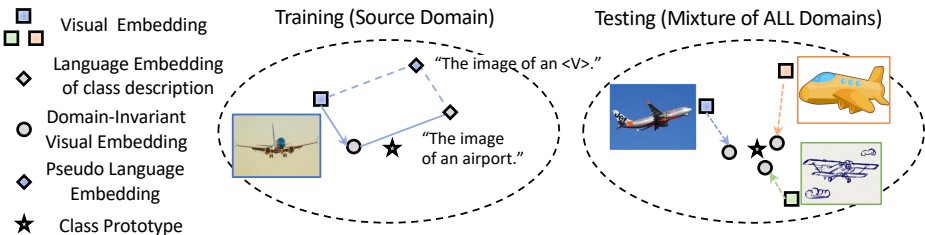

Figure 1: For domain generalization without prior (*i.e.*, domain description, visual example), we propose to learn domain-invariant visual embedding on the joint vision-language embedding space. During training, we estimate domain-invariant representation guided by language embeddings. For test instances of mixed domains (color-coded), we directly predict the domain-invariant part from visual embeddings, and compare it with prototypes learned in source domain for classification.

remove the domain-specific counterpart from the original visual embedding. Notably, as the class description is discriminative, the language embedding of class is naturally domain-invariant.

In this paper, as shown in Fig. 1, we proposed to use language embeddings to guide the learning of domain-invariant representation for each visual embedding. For each class, we treat its language embedding as a domain-invariant anchor. For each image, following the idea of textural inversion (Gal et al., 2022), we propose to Word the Image, *i.e.*, projecting its visual embedding to the token feature of a pseudo word, with the same prompt as the class description to extract pseudo language embedding. Then, the difference with anchor is used to indicate the domain-specific information. As a "global direction" can be aligned across visual and language embeddings in the joint embedding space (Dunlap et al., 2023), we can disentangle the Domain-Invariant visual embedding.

Through experimental validation, our WIDIn is generalizable to vision-language embedding space obtained by either 1) joint large-scale pre-training such as CLIP (Radford et al., 2021) or 2) learning to align two uni-modal models, *e.g.*, MoCo (He et al., 2020) and BERT (Devlin et al., 2018), on data from the single domain. In this way, our approach provides a general solution and can be intuitively extended to optimize the vision and language encoders pre-trained from different datasets and objectives individually. Furthermore, we note the domain-specific information can also be bias caused by the imbalance of training data distribution. Specifically, in long-tail image classification, the pre-trained model may always favor the major classes with a lot of training data, leading to less acceptable accuracy on minor classes. Thus, our approach can be naturally applied on long-tail benchmark datasets to demonstrate its benefits. Our contributions are summarized as follows:

- We propose a domain-invariant representation learning strategy, to extend the systems trained on a single domain to target domains without any prior such as language description and visual data.

- We design WIDIn to first model the domain-specific counterpart as the difference between language embeddings extracted from the class description and the worded images, which is then used to disentangle domain-invariant visual embedding. Meanwhile, our WIDIn can be built on top of vision encoders & language encoders which are trained either jointly or separately.

- We experimentally demonstrate the benefit of our approach on two datasets for domain generalization. We also extend our method to challenging long tail problem and prove the effectivenss on two datasets with high imbalance ratios.

## 2 RELATED WORK

**Debiased Feature Learning** has been critical to learn a robust algorithm (Torralba & Efros, 2011) across different scenarios. In long-tail classification, the bias is caused by imbalance of data distribution and the accuracy on minor classes with limited training data is unacceptable. To mitigate the accuracy gap between minor classes and the major classes counterpart, the approaches based on data re-sampling (Geirhos et al., 2019), training curriculum design Zemel et al. (2013), loss adjustment (Kang et al., 2021; Menon et al., 2020; Lin et al., 2017) are investigated. Besides, the bias can also be caused by the absence of diverse domains (Wang & Deng, 2018) and environments (Creager et al., 2021). Then, to remove such class-irrelevant features which contain domain and instance-

specific information, recent approaches has applied feature disentanglement (Misra et al., 2016), adversarial training (Hoffman et al., 2018), and invariant regularizer (Arjovsky et al., 2019).

**Multimodal Representation** aims to combine the semantic meaning from different modalities, *e.g.*, text and image. With the success of self-attention (Vaswani et al., 2017) and self-supervision such as masked modelling (Devlin et al., 2018; He et al., 2022), the joint encoders (Sun et al., 2019; Tan & Bansal, 2019) are proposed, which take the concatenation of text and images as input and model their correlation automatically. Recently, CLIP (Radford et al., 2021) proposes to set encoders for each modality individually and optimize the networks through contrastive learning (Chen et al., 2020; He et al., 2020). In this way, all extracted representation reside in the common space to be compared directly and used for down-stream application. Recently, Zhai et al. (2022) proposes to transform the pre-trained visual embedding space into the multi-modal embedding space by aligning the language embeddings with the image embeddings extracted by the pre-trained encoder. However, the visual representation is still biased (Dunlap et al., 2022; Yang et al., 2023), and additional update via finetuning (Gao et al., 2021) or prompting (Zhou et al., 2022) is required.

**Domain Adaptation & Generalization** aims to mitigate the performance gap caused by domain shift. To reduce the dependence on label annotation in the supervised adaptation (Motiian et al., 2017), the semi- and weakly-supervised setup are studied (Saito et al., 2019; Baek et al., 2020) while the self-supervision is also useful (Xu et al., 2019). However, all of them depends on additional training data of target domains, instead, we focus on domain generalization (Zhou et al., 2021) where only the data from source domain is used for training. Then, with the assistance of language description, the generative approaches (Wang et al., 2020a; Dunlap et al., 2022) synthesize features across domains. However, the priors of new domains are still needed. Recently, multi-modal prompting, *e.g.*, CoCoOp (Zhou et al., 2022), is generalizable to unseen domains by predicting classifier weights for each test image dynamically. However, it is computational expensive at test time and does not improve the quality of visual embedding. In contrast, our framework is trained one single domain without any other prior and aims to disentangle the domain-invariant representation which is cheap at inference and robust to different domains.

## 3 BACKGROUND

### 3.1 DOMAIN GENERALIZATION

Domain generalization applied a model learned on the source domain to the target domains without any prior. Specifically, the train set $\mathcal{D}_s = \{(c, \mathbf{x})\}$ is from one single domain where $c \in \mathcal{C}$ is the class name for image $\mathbf{x}$. At inference, the model is directly evaluated on the test data from various domains. Formally, the test set is $\mathcal{D}_t = \left( \cup_{i=1}^N \mathcal{D}_{t(i)} \right) \cup \mathcal{D}_s^v$ where $\mathcal{D}_{t(i)}$ and $\mathcal{D}_s^v$ are the test data from target domain $i$ and source domain respectively. For the convenience of description, we set $\mathcal{D}_s^v = \mathcal{D}_{t(0)}$ and then $\mathcal{D}_t = \cup_{i=0}^N \mathcal{D}_{t(i)}$. We note that no prior such as domain label/split is provided for test data. For fair comparison, we calculate the accuracy for each domain $\mathcal{D}_{t(i)}$.

As shown in Fig. 2, for each class $c$, we only use the class name and refer to the default template in CLIP (Radford et al., 2021), *i.e.*, [The image of a/an], to build the general description $l_c$ for all of its images (*e.g.*, the image of an airplane). Then, we extract the language embedding $\mathbf{t}_c = \mathbf{F}_T(l_c) \in \mathcal{R}^d$ where $\mathbf{F}_T$ is the language encoder and $d$ the feature dimension. For the convenience of description, we use $\mathbf{x} \in \mathcal{R}^d$ to indicate the visual embedding extracted by the vision encoder $\mathbf{F}_V$.

### 3.2 ANALYSIS: DOMAIN SHIFT AS DESCRIPTION-DEPENDENT GLOBAL DIRECTION

LADS (Dunlap et al., 2023) proposes to use domain knowledge in CLIP to synthesis visual embeddings for target domains. However, the domain names are required to build description, such as "the clipart of a dog". Without loss of generality, we assume the target domain $t$ and, for each class $c$, the language embeddings of class description in source and target domains are $\mathbf{t}_c^{(s)}$ and $\mathbf{t}_c^{(t)}$ respectively.

Then, LADS assumes a "global direction" of the domain shift in the joint vision-language embedding space. In other words, for each class $c$, the shift direction between visual embeddings of two domains can be indicated by the difference of related language embeddings, *i.e.*,

$$\mathbf{x} - \hat{\mathbf{x}} = k(\mathbf{t}_c^{(s)} - \mathbf{t}_c^{(t)}) \tag{1}$$

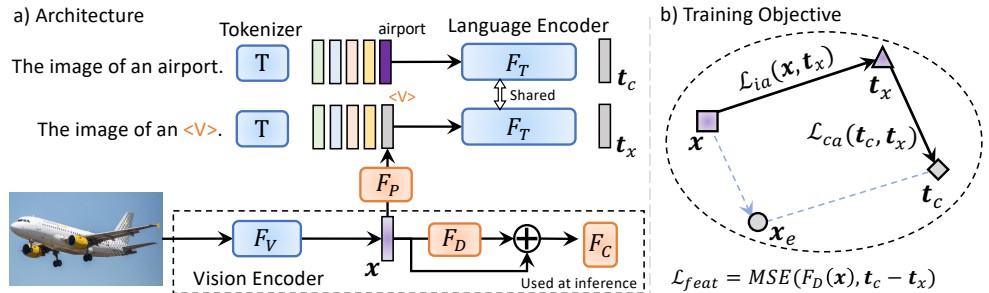

Figure 2: (a) For visual embedding $\mathbf{x}$, we project it to token <V> and extract the pseudo language embedding $\mathbf{t}_x$. We also extract the language embedding $\mathbf{t}_c$ of class $c$ as domain-invariant anchor. At test time, only the modules in dashed box are used. (b) During training, we estimate the domain-specific component as $\mathbf{t}_x - \mathbf{t}_c$ so that the domain-invariant visual embedding $\mathbf{x}_e$ can be disentangled. We apply alignment of $\mathbf{t}_x$ with embeddings $(\mathbf{x}, \mathbf{t}_c)$ at instance- $(\mathcal{L}_{ia})$ and class-level $(\mathcal{L}_{ca})$ respectively, and use residual learning to obtain $\mathbf{x}_e$ by predicting $\mathbf{t}_x - \mathbf{t}_c$. Only modules in red is trained.

where $k$ is a scalar, and the visual embedding $\hat{\mathbf{x}}$ corresponds to $\mathbf{x}$ in domain $s$ but comes from domain $t$. Based on this assumption, LADS proposes to train a feature augmentation network to synthesize $\hat{\mathbf{x}}$ in target domain from $\mathbf{x} \in \mathcal{D}_s$ under the assumption of global direction. However, we found a baseline with $k = 1$ fixed has already provided non-trivial gain (detailed in Supp.) and the gain introduced by the learnable feature augmentation is not significant. Furthermore, even though the augmented feature extends the ability of model across domains, the global direction still ignores the variability across instances where the mapping between $\mathbf{x}$ and $\hat{\mathbf{x}}$ is not always optimal.

Instead, we propose to adaptively model the domain-specific information for each instance, *i.e.*, the gap between visual embedding and domain-invariant visual embedding, instructed by language description. As shown by the textual inversion in image generation (Gal et al., 2022), the word token projected from the image still contains both domain-specific and domain-invariant information. Since language embeddings of classes $\{\mathbf{t}_c\}_{c \in \mathcal{C}}$ can be used as domain-invariant anchors (Radford et al., 2021), for each $(\mathbf{x}, c)$, we propose to compare the pseudo language embedding extracted from the projected token with $\mathbf{t}_c$ to estimate the domain-specific counterpart such that the domain-invariant component can be disentangled via "global direction" and is robust to various domains.

Notably, though we build the framework upon the "global direction" in joint vision-language embedding space, the space can be learned via either large-scale multi-modal pre-training (*e.g.*, CLIP) or aligning two uni-modal models (*e.g.*, BERT and MoCo) on data in $\mathcal{D}_s$ under contrastive learning. Below we first explain our framework in Sec. 4 and then detail the experimental studies in Sec. 5.

## 4 WORDING IMAGE FOR DOMAIN-INVARIANT REPRESENTATION LEARNING

We explain the network architecture in Sec. 4.1, and detail the network training in Sec. 4.2.

### 4.1 ARCHITECTURE

As shown in Fig. 2(a), we obtain the domain-invariant representation guided by language embeddings, *i.e.*, comparing the language embeddings extracted from the descriptions with class name and the projected word token <V>. Specifically, we define a two-layer MLP (LeCun et al., 2015) as the projector $\mathbf{F}_P$ to project the visual embedding to the token embedding. Then, we build the input to $\mathbf{F}_T$ with the prompt [The image of an <V>] and extract pseudo language embedding $\mathbf{t}_x$, *i.e.*,

$$\mathbf{t}_x = \mathbf{F}_T\big(\text{Concatenate}[\mathbf{T}(\text{The image of an}), \mathbf{F}_P(\mathbf{x})]\big) \tag{2}$$

where $\mathbf{T}$ denotes the function to tokenize the input sentence and retrieve token embeddings of dimension $d$, $\mathbf{F}_P(\mathbf{x}) \in \mathcal{R}^d$ is the embedding indicated by <V>. As $\mathbf{t}_x$ is extracted from the visual embedding $\mathbf{x}$, we assume $\mathbf{t}_x$ still exhibits both domain-invariant and domain-specific information.

For each $\mathbf{x}$ of class $c$, we use the same prompt and language encoder to extract both $\mathbf{t}_x$ and $\mathbf{t}_c$. As we treat $\mathbf{t}_c$ as the domain-invariant anchor, we use their comparison, *i.e.*, $\mathbf{t}_x - \mathbf{t}_c$, to estimate the domain-

specific information. Based on the observation where the shift between language embeddings can be generalizable to the visual embeddings in the joint vision-language embedding space, we can then obtain the domain-invariant visual embedding $\mathbf{x}_e$ of $\mathbf{x}$, *i.e.*,

$$\mathbf{x}_e = \mathbf{x} - (\mathbf{t}_x - \mathbf{t}_c). \tag{3}$$

and input it to classifier $\mathbf{F}_C$ for recognition. Meanwhile, we learn a module $F_D$ to predict domain-invariant feature $\mathbf{x}'_e$ for each visual embedding at testing. To stabilize the network training, all three embeddings $(\mathbf{x}, \mathbf{t}_c, \mathbf{t}_x)$ are pre-processed by $l_2$-normalization, and we follow the idea of residual learning (He et al., 2016) to train $F_D$ as a disentangler to predict the feature $\mathbf{t}_c - \mathbf{t}_x$. The weights in classifier $\mathbf{F}_C$ are $l_2$-normalized and, without loss of generality, are referred as class prototypes.

We note our framework is different from conventional prompting algorithm (Jia et al., 2022) where the images are used to provide context for down-stream task application. Instead, for each image, by mapping the global visual embedding to the word token embedding, we aim to extract the embedding which can facilitate the domain-invariant representation learning. As a result, it should be both 1) aligned with the original visual embedding $\mathbf{x}$ and 2) associated the domain-invariant anchor $\mathbf{t}_c$.

## 4.2 TRAINING OBJECTIVE AND SCHEDULE

As shown in Fig. 2(b), for each visual instance, the pseudo language embedding should both 1) be comparable with the language embeddings of classes and 2) contains the information of original visual embedding. As such, we consider both the class-level alignment with the language embeddings $\{\mathbf{t}_c\}_{c \in \mathcal{C}}$ and the instance-level alignment with the original visual embeddings. We refer to the conceptually simple contrastive learning for the alignment constraint and explain it below.

**Contrastive learning** (CT) aims to group the samples with high correlation in a self-supervised manner, and treats different views of the same instance as correlated (positive). Then, the negative pairs are built by sampling views from other instances. Formally, given a instance $\mathbf{x}$ with its paired view $\mathbf{x}'$, the views from other instances, *i.e.*, $\mathcal{B}_n$, are used to build negative pairs. Then, the training objective is to minimize the InfoNCE loss (Oord et al., 2018),

$$\mathcal{L}_{CT}(\mathbf{x}, \mathbf{x}', \mathcal{B}_n) = -\log \frac{\exp(\mathbf{x} \cdot \mathbf{x}'/\tau)}{\exp(\mathbf{x} \cdot \mathbf{x}'/\tau) + \sum_{\mathbf{m} \in \mathcal{B}_n} \exp(\mathbf{x} \cdot \mathbf{m}/\tau)}. \tag{4}$$

where $(\mathbf{x}, \mathbf{x}', \mathbf{m})$ are pre-processed by $l_2$-normalization and $\tau$ is a hyperparameter to rescale the affinity scores. In this way, CT is performing pair-wise comparison. Then, the supervised contrastive learning (SupCT) is instead introducing the class labels where the images of the same class are treated as positive while the rest are used to build negative pairs.

**Alignment of pseudo language embedding.** For class-level alignment, we follow SupCT to contrast each pseudo language embedding with all language embeddings of classes $\{\mathbf{t}_c\}_{c \in \mathcal{C}}$, which is equivalent to cross-entropy (CE) loss in fully-supervision (LeCun et al., 2015), *i.e.*, $\mathcal{L}_{ca} = \mathcal{L}_{CT}(\mathbf{t}_x, \mathbf{t}_c, \{\mathbf{t}_{c'}\}_{c' \neq c})$. Then, at the instance-level, visual embedding $\mathbf{x}$ and its pseudo language embedding $\mathbf{t}_x$ serve as *two different views* to build a positive pair where the other embeddings are negative samples, *i.e.*, $\mathcal{L}_{ia} = \mathcal{L}_{CT}(\mathbf{t}_x, \mathbf{x}, \{\mathbf{x}_n\})$ where $\{\mathbf{x}_n\}$ are embeddings of other instances. Then, given $\mathbf{t}_c - \mathbf{t}_x$ as the groundtruth, we train $\mathbf{F}_D$ to predict it and directly set the Mean Squared Error (MSE) as the loss $\mathcal{L}_{feat}$. In addition, we use CE loss to train the classifier $\mathbf{F}_C$.

**Implementation.** To avoid training collapse, we schedule the entire training framework into two steps. Firstly, we only train the projector $F_P$ under alignment losses $\mathcal{L}_{ia} + \mathcal{L}_{ca}$. Then, we fix the pre-learned $\mathbf{F}_P$, and train the feature disentangler $F_D$ and the classifier $F_C$. We set batch size as 64 and the experiments can be realized on one TiTAN RTX GPU. More details can be found in Supp.

## 5 EXPERIMENT

Here the experiment setup is first explained, to validate the effect of WIDIn in joint vision-language embedding space learned by 1) large-scale pre-training and 2) finetuning uni-modal models.

**Dataset.** 1) **CUB-Painting** consists of 200 finegrained classes of bird species from two domains constructed upon CUB-200-2011 (Wah et al., 2011) and CUB-200-Paintings (Wang et al., 2020b). The former one (source) contains the pictures taken from natural world, while the latter one (target)

Table 1: Performance comparison of domain generalization.

| Approach | CUB-Painting | | | DomainNetMini | | |
|---|---|---|---|---|---|---|
| | Src. | Tar. | Avg. | Src. | Tar. | Avg. |
| Zero-Shot (Radford et al., 2021) | 60.34 | 52.84 | 56.59 | 93.49 | 95.94 | 94.72 |
| Zero-Shot* (Radford et al., 2021) | 61.93 | 54.38 | 58.16 | 93.24 | **96.01** | 94.62 |
| WiSE-LD (Wortsman et al., 2022) | 81.74 | 64.80 | 73.27 | 95.19 | 93.68 | 94.44 |
| VQGAN+CLIP (Crowson et al., 2022) | - | - | - | 95.54 | 93.83 | 95.27 |
| LADS (Dunlap et al., 2022) | 86.14 | 66.18 | 76.16 | 95.33 | 95.21 | 95.27 |
| Linear Clf. | 85.91 | 64.33 | 75.12 | 95.03 | 93.75 | 94.39 |
| MLP Clf. | 86.06 | 62.32 | 74.19 | 96.33 | 92.47 | 94.40 |
| WIDIn (Ours) | **87.47** | **68.76** | **78.12** | **97.08** | **96.01** | **96.75** |

\*: include domain name in extracting language embedding for each class.

is in the painting style. 2) **DomainNetMini** (Tan et al., 2020) is a subset of DomainNet (Peng et al., 2019) and contains 40 classes from four different domains, *i.e.*, sketch, real, clipart and painting. We follow the commonly used evaluation setup (Dunlap et al., 2023) where the model is only trained on "sketch" domain (source) and the rest three domains (target) are unseen during training.

**Evaluation Metric**. We report the classification accuracy on the source (Src.) and target (Tar.) domains respectively, and use the average (Avg.) to measure the performance on all domains.

**Generalization on large-scale pre-trained embedding space.** Following Dunlap et al. (2023), we learn the domain-invariant representation from visual embeddings in the space learned by CLIP (Radford et al., 2021), *i.e.*, we use the CLIP vision encoder to extract visual embeddings and then use CLIP language encoder for feature disentanglement. For fair comparison, as summarized in Table 1, we use the ViT-$L_{14}$ as the backbone and more details can be found in the Supp.

CLIP has shown remarkable ability to generalize across different tasks. However, on CUB-Painting, using language embeddings of classes as centers (*i.e.*, Zero-Shot) cannot clearly identify the semantic details for fine-grained objects while learning a linear classifier (Linear Clf.) can improve the performance significantly on the source domain. [*Stronger classifier results in larger performance gap*] However, the simple Linear Clf. on the CLIP embedding space has resulted in clear performance gap between Src. and Tar. On DomainNetMini, it even underperforms the zero-shot counterparts on Tar. In addition, by employing a stronger classifier, MLP Clf., which includes more parametes in the classifier but is only trained by minimizing cross-entropy loss, the performance gap is enlarged and the accuracy on Tar drops further. [*Domain-invariant representation improves the accuracy consistently*] Instead, though both LADS and our WIDIn are built upon the Linear Clf. baseline, our WIDIn clearly model the domain-invariant representation and can outperform LADS and all other baselines consistently on all cases but does not require domain prior.

Table 2: Performance comparison of domain generalization on space learned by aligning.

| (a) Full-Supervision (LeCun et al., 2015) | | | | | | (b) MoCo (He et al., 2020) | | | | |
|---|---|---|---|---|---|---|---|---|---|---|
| | CUB-Painting | | DomainNetMini | | | | CUB-Painting | | DomainNetMini | |
| | Src. | Tar. | Src. | Tar. | | | Src. | Tar. | Src. | Tar. |
| Linear Clf. | 83.28 | 57.47 | 86.94 | 56.41 | | Linear Clf. | 78.10 | 54.41 | 83.79 | 46.07 |
| CLIP-Text | 88.33 | 59.42 | 91.99 | 63.35 | | CLIP-Text | **88.44** | **59.18** | **91.55** | **61.44** |
| BERT | **90.97** | **59.81** | **93.64** | **64.75** | | BERT | 85.12 | 57.85 | 90.81 | 59.51 |

**Generalization on aligned uni-modal models.** From Eq. 3, the property of "global direction" exhibited in the joint vision-language embedding space is essential for transferring the domain-specific information estimated by language embeddings to the visual parts. Thus, to apply our approach on uni-modal models trained on irrelevant datasets, *e.g.*, MoCo (He et al., 2020) and BERT (Devlin et al., 2018) that are pre-trained on ImageNet Deng et al. (2009) and Engligh Wikipedia (Zhu et al., 2015) individually, additional training is needed to align the embedding space.

[*Model selection & preparation*] Without loss of generality, we fix the language encoder and finetune the vision encoder by minimizing the multi-modal contrastive loss (Radford et al., 2021) on the

train set $\mathcal{D}_s$ while a linear layer can be added on top of the vision encoder to align the embedding dimension if necessary. Then, our approach WIDIn can be applied. As shown in Table 2, we select the language encoder learned in CLIP (Radford et al., 2021) (CLIP-Text) and BERT (Devlin et al., 2018). Both CLIP-Text and BERT are pre-trained in a self-supervised manner, where the former one is trained by contrasting paired visual embeddings while the latter one is purely trained on web text. For image encoder, we select ViT-B$_{16}$ which is trained via full-supervision or self-supervision (*i.e.*, MoCo), on Imagenet. More experimental evaluation can be found in the Supp.

As a fair comparison, for the baseline Linear Clf., the backbone parameters are also updated jointly with the classifier during the training. As Full-Supervision train the encoder for classification in the pre-training, the extracted features are more discriminative, and the Linear Clf. by MoCo underperforms the former one. However, by using either CLIP-Text or BERT as the text encoder in our WIDIn framework, the performance on all datasets are improved consistently.

Meanwhile, we observed that accompanying BERT with encoders in Full-Supervision works better while CLIP-Text can facilitates more on MoCo encoder. Our assumption is that the uni-modal models whose training objectives are complementary to each other can achieve better performance.

- As the Full Supervision trains the vision encoder by minimizing cross-entropy loss, the embeddings may focus more on global semantic meaning. Then, as BERT is trained with multiple tasks including word prediction that acquires local semantic modeling, it can better locate the domain-specific component exhibited in the visual embedding to facilitate the feature disentanglement.
- On the other hand, MoCo is essentially performing pair-wise comparison across instance embeddings directly and can contain more local details in the extracted embeddings. Then, as CLIP-Text is learned under the contrastive objective to cluster features with high semantic similarity, the language embeddings of classes can serve as desired domain-invariant anchor so that the domain-specific information for each instance can be better represented.

In sum, as learning a large-scale vision-language model is expensive in both data collection and network training, and such a giant model does not always outperform a smaller one specifically trained in one modality, *e.g.*, the vision encoder in CLIP underperforms the backbone by MAE (He et al., 2022) for object detection, we hope WIDIn provides more insights for aligning uni-modal models for different downstream applications.

## 6 DISCUSSION

### 6.1 ABLATION STUDY

Table 3: Ablation study on embedding types for (a) object and (b) domain recognition.

| (a) Object Recognition ($\uparrow$) | | | | | | (b) Domain Recognition ($\downarrow$) | | |
|---|---|---|---|---|---|---|---|---|
| | CUB-Painting | | DomainNetMini | | | | CUB-Painting | DomainNetMini |
| | Src. | Tar. | Src. | Tar. | | | | |
| Original | 85.91 | 64.33 | 95.03 | 93.75 | | Original | 90.85 | 76.22 |
| Domain-Specific | 51.78 | 36.82 | 37.16 | 30.97 | | Domain-Specific | 97.31 | 88.91 |
| Domain-Invariant | **87.47** | **68.76** | **97.08** | **96.01** | | Domain-Invariant | **75.30** | **48.14** |

**Ablation study on domain-invariant representation.** Under the guidance of language embedding, the domain-invariant representation $\mathbf{x}'_e$ is predicted from the visual embedding $\mathbf{x}$ in WIDIn and both of them reside in the same embedding space. As the embedding is predicted via residual learning, the output of feature disentangler, *i.e.*, $-F_D(\mathbf{x})$, essentially contains domain-specific information. Thus, we provide the ablation study on these three embedding types for object category recognition and domain recognition.

For the domain recognition, we train on the embeddings from all domains to learn a linear domain classifier, *i.e.*, {natural and painting} for CUB-Painting and {sketch, real, clipart and painting} for DomainNetMini. As the origin dataset split does not have training samples from target domains, we use the union of train set $\mathcal{D}_s$ and half of test set per target domain to train the domain classifier (necessary sub-sampling is applied to balance the train set per domain), and then report the average of per-class accuracy on the rest test data.

Table 4: Robustness to prompts.

| | Src. | Tar. |
|---|---|---|
| Random | 87.49 | 68.84 |
| Aggregated | 88.51 | 70.01 |
| Misaligned | 87.13 | 68.20 |

Table 5: Ablation on alignment.

| | Src. | Tar. |
|---|---|---|
| None | 86.42 | 67.12 |
| SupCT | 87.11 | 67.54 |
| CT | **87.47** | **68.76** |

Table 6: Accuracy of pseudo language embedding

| Dataset | Src. | Tar. |
|---|---|---|
| CUB-Painting | 84.43 | 59.76 |
| DomainNetMini | 97.00 | 93.86 |

From Table. 3, as the domain-specific information mainly contains the clues of different domains, it can be used to recognize the domain correctly but is less effective in recognizing the object classes. In contrast, as the domain-invariant embedding is obtained by deducting the domain-specific counterpart from the original embedding, it is more robust in object recognition across different domains.

**Robustness to prompts.** WIDIn can be applied to different prompts and training among a mixture of prompts can improve the accuracy further. In addition to [The image of an <V>], we consider other two prompts, [The photo of an <V>] and [<V> in the scene], and use CUB-Painting for validation. Comparing Table 1 & 4, when randomly selecting prompts for each image during training, the resulted score (*Random*) is similar. Then, we consider aggregating the embeddings extracted from multiple prompts. Formally, we gather outputs $\{\mathbf{t}_c^{(i)}\}_{i=1}^N$ and $\{\mathbf{t}_x^{(i)}\}_{i=1}^N$ where $i$ indexes the prompt and $N = 3$. In this way, we treat $\bar{\mathbf{t}}_c = \frac{1}{N}\sum_i \mathbf{t}_c^{(i)}$ and $\bar{\mathbf{t}}_x = \frac{1}{N}\sum_i \mathbf{t}_x^{(i)}$ as the domain-invariant anchor and the pseudo language embedding for disentanglement. As more embeddings are used to smooth the noise in each single prompts, the score is improved (*Aggregated*). Meanwhile, to correctly estimate the domain-specific feature, we note the prompt used to extract language embeddings from class description and projected word token at each time should be the same. Otherwise, the difference between prompts may introduce more noise (*e.g.*, $\mathbf{t}_x^{(2)} - \mathbf{t}_c^{(1)}$) and result in slight drop (*Misaligned*).

**Alignment between visual embedding and pseudo language embedding** is achieved by minimizing the CT. As an alternative, we apply SupCT. As shown in Table 5, since we study on the pre-trained embedding space of CLIP, applying no alignment (None) $\mathbf{x}$ and $\mathbf{t}_x$ can still achieve reasonable gain while applying either SupCT or CT still helps. However, since SupCT includes class-labels to determine the positive & negative samples and inevitably lose the instance-level details, the model cannot then properly estimate the domain-specific between language embedding of class description and the pseudo language embedding, resulting in limited gain in Tar.

**Generalization ability of pseudo language embedding.** As the pseudo language embeddings is trained under alignment constraint, it also be used for evaluation. However, as we fix the language encoder during training, the performance is highly depends on quality of pre-learned language embeddings. As summarized in Table 6, by comparing with Table 1, though the accuracy is improved against zero-shot baseline. the Tar. accuracy still under-performs the Linear Clf. baseline in fine-grained classification. Instead, the pre-trained visual encoder can better preserve the tiny details. In this way, our approach does not forget the essential details in visual inputs but just remove the unnecessary domain-specific component such as background and domain. Furthermore, the computational complexity is much higher during inference as the text encoder is additionally required.

**Estimation of domain-specific feature**. As a naive baseline, we can ignore the pseudo language embedding and directly represent the domain-specific information as the difference between visual embedding and language embedding of its class, *i.e.*, $\mathbf{x} - \mathbf{t}_c$. However, the performance is sub-optimal (CUB-Painting, Src. 63.77 and Tar. 48.98). After all, even though the CLIP has been pre-trained to minimize the contrastive loss between the paired image and language description, the embeddings extracted from the two modalities may still be distant from each other. In other words, $\mathbf{x} - \mathbf{t}_c$ additionaly contains the instance-agnostic gap between embeddings of different modalities and it is difficult to learn with limited data.

## 6.2 EXTENSION TO LONG-TAIL IMAGE RECOGNITION

Besides the domain defined by background and environment in the images, the domain can also be generalized to the statistical distribution of training data where the domain bias can be caused by distribution imbalance. In detail, for long-tail learning where the frequency of training data per class varies from each other significantly, the model may favor the major classes with sufficient training

Table 7: Performance comparison of long-tail classification.

| Approach | Backbone | ImageNet-LT | | | | iNaturalist |
| --- | --- | --- | --- | --- | --- | --- |
| | | Few | Med | Many | Overall | Overall |
| LWS (Kang et al., 2020) | RN50* | 29.30 | 45.20 | 57.10 | 47.70 | 65.90 |
| LogitAdj (Menon et al., 2020) | RN50* | 49.94 | 52.32 | 50.06 | 51.13 | 68.44 |
| TSC (Li et al., 2022) | RN50* | 30.40 | 49.70 | **63.50** | 52.40 | 69.70 |
| NCM (Kang et al., 2020) | RN50^ | 31.10 | 46.60 | 58.90 | 49.20 | 65.30 |
| cRT (Kang et al., 2020) | RN50^ | 27.80 | 47.20 | 63.30 | 50.80 | 69.90 |
| $\tau$-normalized (Kang et al., 2020) | RN50^ | 33.80 | 48.40 | 60.90 | 51.20 | 71.20 |
| LWS (Kang et al., 2020) | RN50^ | 31.80 | 48.60 | 62.20 | 51.50 | 71.00 |
| Zero-Shot | RN50† | 52.92 | 57.78 | 58.13 | 56.61 | 3.41 |
| Baseline | RN50† | 53.05 | 60.24 | 59.17 | 58.85 | 71.85 |
| WIDIn (Ours) | RN50† | **55.12** | **61.26** | 60.56 | **60.55** | **73.20** |
| Zero-Shot | ViT-B16† | 63.63 | 66.43 | 66.86 | 66.71 | 4.16 |
| Baseline | ViT-B16† | 67.84 | 67.72 | 68.23 | 67.95 | 73.72 |
| WIDIn (Ours) | ViT-B16† | **68.91** | **71.69** | **72.07** | **71.93** | **75.45** |

^: Initialized with CLIP (Radford et al., 2021) and jointly trained. Results reported by Tian et al. (2022).
*: Trained from scratch. †: *Fixed* and initialized with CLIP (Radford et al., 2021). More details can be found in Supp.

data and the accuracy on minor classes with limited data is less acceptable. Thus, how to balance the classification accuracy on all classes is also important.

As shown in Table 7, we evaluate our approach on two benchmark datasets, **ImageNet-LT** (Liu et al., 2019) and **iNaturalist** (iNatrualist, 2018), where the number of training samples per class ranges in [5,1280] and [2,1000] respectively. According to the number of training data of each class, we split the classes into three groups, Many ($\geq 100$ samples), Medium (Med, 20~100 samples) and Few ($\leq 20$ samples). Then, for evaluation, we report the performance on all classes as well as the accuracy over the groups. Meanwhile, for our approach, we report the performance with both RN50 and ViT-B$_{16}$ as the backbone and use pre-trained CLIP model as initialization for fair comparison.

Though CLIP shows strong zero-shot accuracy on ImageNet-LT, it does not perform well on fine-grained classes in iNaturalist. We note that most approaches compared below requires to update the vision encoder parameter during training to achieve high score, which is computational expensive. Instead, for the efficiency of experiments, we fix the encoders but add class-specific margins (Menon et al., 2020) in the calculation of cross-entropy loss (Baseline) and WIDIn can still achieve consistent performance gain. We believe that our approach can further improve the performance when the vision encoder is also jointly trained with the projector.

**Limitations and societal impacts**. As introduced in Sec. 4, we estimate the domain-specific information by comparing embeddings extracted from the language encoder. Therefore, the estimation relies on the quality of the language model. Also, WIDIn takes the output corresponding to `<CLS>` token as the language embeddings, which does not intuitively apply to autoregressive language models where no `<CLS>` token is learned. We will leave it for future work. To the best of our knowledge, as our work is purely an algorithm for learning domain-invariant features, we haven not found any negative societal impact.

## 7 CONCLUSION

In this paper, we investigate the problem of domain generalization, and propose the WIDIn that is free from any prior on target domains. For each image, we first word the image by projecting the visual embedding to a word token embedding. We then use the difference between language embeddings extracted from worded image and class description to estimate domain-specific feature. Based on the "global direction" property in multi-modal embedding space, we can disentangle the domain-invariant representation from the original visual embeddings, which is robust to different domains. The experimental study demonstrates the effect of WIDIn on domain generalization as well as long-tail learning and can be intuitively applied to aggregate the knowledge from visual encoders and language encoders which are learned on irrelevant datasets separately. Detailed ablation studies are used to justify components of our approach design.

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

# A BASELINE BY GLOBAL DIRECTION

## A.1 BASELINE DETAILS IN DOMAIN GENERALIZATION

As discussed in Sec. 3.2, LADS (Dunlap et al., 2023) assumes a global direction of domain shift exhibited in both language embeddings and visual embeddings. To this end, they propose the direction loss to train a feature augmentation network and synthesize the features of target domains. Though significant gain (from 64.33 to 66.18 on CUB-Painting) is reached on target domains, one important baseline is missed to highlight the importance of the global direction and justify the importance of training a neural network.

Instead of training a feature augmentation network under the constraint of direction loss, a baseline can directly set $k = 1$ and use the language embeddings to obtain new features $\hat{\mathbf{x}}$, *i.e.*, $\hat{\mathbf{x}} = \mathbf{x} + \mathbf{t}_c^{(t)} - \mathbf{t}_c^{(s)}$. However, as the extracted embeddings are of different norms, the direction obtained from the language embeddings, *i.e.*, $\mathbf{t}_c^{(t)} - \mathbf{t}_c^{(s)}$, is not theoretically guaranteed to be consistent with the ground truth global direction. As such, to maintain the stability of the network training, we pre-process all of the embeddings $(\mathbf{x}, \mathbf{t}_c^{(t)}, \mathbf{t}_c^{(s)})$ by $l_2$-normalization.

Then, we directly use $\mathbf{x}$ and $\hat{\mathbf{x}}$ after normalization to train a linear classifier where the weights in the classifier are also normalized. In this way, we essentially use cosine similarity as the logits. For this baseline approach on CUB-Painting, we obtain the performance 85.93 (Src.) and 65.21 (Tar.), where the performance gain on the target domain is non-trivial. In other words, by comparing with this baseline, training an additional feature augmentation network is still useful but the introduced gain is then less than 1%. Nevertheless, both this baseline and LADS demonstrate the knowledge of domain exhibited in multi-modal embedding space and the importance of global direction.

In addition, following the normalization step to pre-process the visual embeddings and language embeddings, for our approach WIDIn, in Eq. 3, all of the embeddings $(\mathbf{x}, \mathbf{t}_c, \mathbf{t}_x)$ are also normalized to ensure the stability and proper estimation of the domain-specific information.

## A.2 MOTIVATION & CLARIFICATION OF WIDIN

Based on the observation above, we think the global direction exhibited in the joint vision-language embedding space that is learned through multi-modal contrastive learning is beneficial for domain generalization. Then, to obtain the joint multi-modal embedding space, we consider two cases, 1) CLIP embedding space which is obtained through multi-modal contrastive learning on LAION 400M, and 2) finetuning from two uni-modal models which are trained on irrelevant large-scale uni-modal datasets through multi-modal contrastive learning on the training set in source domain.

**Clarification of large-scale pretrained models**. The large-scale model refers to the models pretrained on any large-scale datasets such as LAION and ImageNet. For domain generalization, using ImageNet-Pretrained model as initialization has been a common practice. However, the performance on target domains is stil less acceptable. Meanwhile, for the vision encoder in CLIP pretrained on LAION, there is still unignorable performance gap between the source domain and target domains. As such, we propose WIDIn, a general framework that can be applied on both cases to learn domain-invariant representation and improve the domain generalization ability. Meanwhile, our experimental validation also shows using a language model such as BERT which is irrelevant with the image model at all can still be helpful for feature disentanglement.

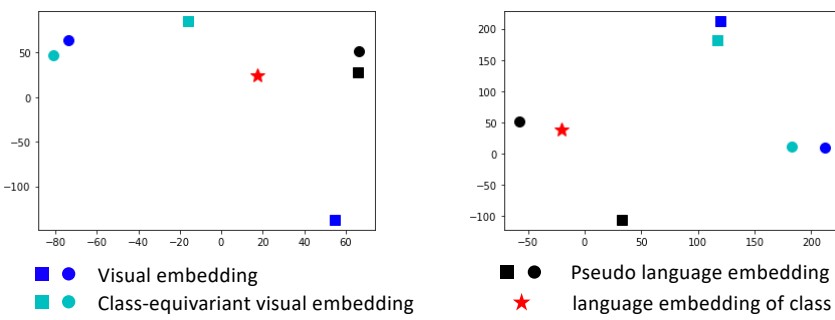

Figure 3: Visualization of embedding of two examples, the embedding types are color coded.

# B  VISUALIZATION & DISCUSSION

## B.1  COMPARISON BETWEEN VISION AND LANGUAGE EMBEDDINGS

From the ablation studies reported in Sec. 6.1, as pseudo language embeddings are learned under both the class-level and instance-level alignment, they can thus be also used for evaluation by comparing with the language embeddings extracted from the class descriptions. Even though the performance is improved, we found that it is still significantly under-perform the linear evaluation baseline trained on the visual embeddings.

As such, we think the distribution of language embeddings and the distribution of visual embeddings may not be perfectly aligned and the visual embeddings are capable to capture tiny details which are important for fine-grained classification but not captured by the language embeddings. Towards this end, from the perspective of comparison, we believe it is unfair to only compare the zero-shot performance with any approach trained on visual embeddings. Instead, the linear evaluation should also be considered and accompanied with the zero-shot baseline for reference.

## B.2  VISUALIZATION

As shown in Fig. 3 below, we provide two visualization examples of the embeddings. Within each example, we plot the visual embeddings, pseudo language embeddings, and the domain-invariant visual embeddings of two instances (the cycle shape denotes the instance from source domain and the square shape denotes the instance from the target domain), as well as the language embeddings of class description. In both examples, comparing with the visual embeddings of the two instances, the domain-invariant visual embeddings are closer to each other. In this way, the classifier learned on the instances of source domain can also perform well on the instances from target domains.

## B.3  GENERALIZATION TO MULTI INSTANCES

Our approach can be naturally generalized to images containing multiple classes, as it leverages the difference between pseudo language embedding and language embedding of caption to model the class-invariant information. For example, given the caption "The image of a cat and a dog" and its paired image, as the original visual embedding has encoded the information from both the cat and the dog, the corresponding pseudo language embedding can still be compared with the language embedding of paired caption to estimate the information of background and domain.

At the same time, to model the domain-invariant information for each object in the image, we can extend the prompt for single concept to the prompt [The image of a $<V_1>$ and a $<V_2>$].

Then, we can follow Carion et al. (2020) to design the projector where the visual embedding can be mapped to different word tokens. As the classes mentioned in the caption is commutative, *e.g.*, "The image of a cat and a dog" is equivalent to "The image of a dog and a cat", and the projector outputs are not ordered. We can also use Hungarian algorithm (Carion et al., 2020) to find the correct matching between them and then facilitate the modeling of class-invariant feature. We will leave the application for images containing multiple classes such as object detection for future work.

Table 8: Ablation study on CUB-Painting for aligning pre-trained Image & Language Encoders

| Architecture | Fully-Supervised (LeCun et al., 2015) | | MoCo (He et al., 2020) | | MAE (He et al., 2022) | | DINO (Caron et al., 2021) | |
|---|---|---|---|---|---|---|---|---|
| | ViT-B$_{16}$ | | ViT-B$_{16}$ | | ViT-B$_{16}$ | | ViT-B$_{16}$ | |
| | Src. | Tar. | Src. | Tar. | Src. | Tar. | Src. | Tar. |
| Linear Clf. | 66.29 | 37.74 | 53.18 | 25.66 | 32.03 | 31.99 | 80.45 | 52.67 |
| CLIP-Text | 70.31 | 35.08 | 75.49 | 39.51 | 54.42 | 48.08 | 81.34 | 53.36 |
| BERT | 75.6 | 41.25 | 72.94 | 35.77 | 51.07 | 42.87 | 81.71 | 54.05 |
| Architecture | RN50 | | RN50 | | - | | ViT-B$_8$ | |
| | Src. | Tar. | Src. | Tar. | - | - | Src. | Tar. |
| Linear Clf. | 50.12 | 22.32 | 49.52 | 14.17 | - | - | 80.98 | 49.66 |
| CLIP-Text | 51.85 | 21.6 | 53.4 | 19.4 | - | - | 82.62 | 52.41 |
| BERT | 68.67 | 26.49 | 49.53 | 14.67 | - | - | 82.62 | 52.84 |

## C  DISCUSSION OF WIDIN ON UNI-MODAL MODELS AFTER ALIGNMENT

Training a joint vision-language embedding space from scratch is computational expensive and how to utilize existing uni-modal model is also important. After all, as demonstrated in Table 1, when adapting a pre-trained model to a downstream tasks, though new knowledge are learned, training a single linear classifier still results in knowledge forgetting. Thus, we hope to mitigate this issue aligning uni-modal models from other modality in our WIDIn framework. However, fusing information from two separately trained uni-modal models should also be of low cost while finetuning the entire network is still computational expensive. As such, below we validate the effect of our approach WIDIn on the joint embedding space which are learned by only updating the feature space.

### C.1  MODEL SELECTION

**Comparison of Vision Encoders**. As shown in Table 8, besides Full-Supervision and MoCo, we also select additional self-supervised image encoders, DINO (He et al., 2022) and MAE (He et al., 2022). Meanwhile, all of the four models are pre-trained on ImageNet (Deng et al., 2009).

- DINO is trained by minimizing the difference of logits by the student encoder and by the teacher encoder in a self-supervised manner. The logits in DINO is obtained by projecting from the features through a linear layer and the projecter layer consisting of 65536 weight vectors. The weight vectors are jointly learned with the entire feature extractor. When we minimize the loss of DINO, for each image instance, we are pushing the features towards the same weight vector. As the number of instances in the train set is much larger than the number of weight vectors, the features from different instances with high similarity will be pushed towards the same class center. As such, the features learned from DINO can be properly clustered.

- Similarly, as the Full Supervision is essentially pushing the image features towards the same class center, the features are clearly trained to be grouped according to the class label.

- MoCo is similar to SimCLR (Chen et al., 2020) and directly compares the features. It is also studied in (Wang & Isola, 2020) that contrastive learning aims to learning a uniform distribution across instances.

- MAE is only trained to reconstructed the masked area and does not directly learns to matching instance features of high similarity.

**Clarification of Language Encoders.** Both CLIP-Text and BERT have different architectures in the published versions. For the convenience of comparison, we only choose the model version whose final the feature dimension is 768, which are the same as the language encoders used in Table. 2. For the convenience of accommodation, we only select the model where a class token <CLS> (Devlin et al., 2018) is learned.

**Implementation Detail**. Specifically, we use the pre-defined data pre-processing function to extract the embeddings, and only train a simple linear layer on top of the pre-extracted embedding, *i.e.*, no data augmentation on the raw images during training. In this way, we can learn a joint image-language embedding space to hold the assumption of the "global direction". Since we are treating

the language embedding as the domain-invariant anchor to instruct the disentanglement of the visual embedding, while learning to map layers for language embeddings based on extremely limited language training data may result in information loss, we only add one linear layer on top of the vision encoders. In other words, we freeze the huge backbone and only apply our approach on the pre-extracted embeddings with necessary pre-processing. We train the layer by aligning the vision embedding with the language embeddings via minimizing contrastive loss (Radford et al., 2021) on the train set from source domain $\mathcal{D}_s$. In this way, we essentially treat the space that the language embeddings reside as the joint vision-language embedding space.

## C.2 EXPERIMENT DISCUSSION

Table 8 summarizes the results on CUB-Painting of WIDIn, along with the performance by learning a simple linear classifier (Linear Clf.) as baseline for reference. For the fairness of comparison, Linear Clf. only train on the pre-extracted embeddings to learn a linear classifier. However, for MAE and MoCo, we observed that it is essential to apply the data augmentation function during the training such that a more robust linear classifier can be learned. Furthermore, we found a batchnorm layer is added, *i.e.*, the extracted visual embeddings of the images first pass the batchnorm layer and the outputs of the batch norm layer are then fed into the classifier, in their official github repos.

**Self-supervision shows less bias to source domain**. Comparing with Linear Clf., WIDIn can consistently improve the accuracy on both source and target domains in most cases. The only exception is the Tar. accuracy using CLIP-Text and the vision encoder pre-trained in a supervised manner. Since the training data of ImageNet is also captured in natural world, the vision encoder may overfit to source domain such that the visual embeddings in target domain have already been distant from those of natural images. As such, the classifier learned from the domain-invariant component in the source domain cannot be generalized to the target domain. Furthermore, as CLIP-Text is trained with contrastive loss while the BERT is pre-trained under multi-task learning, BERT is better in capturing tiny details in the visual data. As such, with the vision encoder trained in a fully-supervised manner, the CLIP-Text can only facilitate the generalization on the original domain.

Meanwhile, though the vision encoder in CLIP still achieve the best performance, we note the pre-train set of CLIP is much larger than ImageNet. And we believe the performance can still be further improved if a stronger pre-trained vision encoder is provided.

**CLIP-Text Vs. BERT**. Through comparison, BERT provides more gain for the vision encoder pretrained by DINO and Full Supervision while CLIP-Text benefits more for the other pre-trained models.

**Generalization over Architectures**: For each vision encoder, we consider both Transformer- and ResNet-based architectures if applicable and available. As ViT-B is of more learnable parameters than RN50 (86M Vs. 23M), ViT-B outperforms RN50 consistently. Meanwhile, we note the benefit of our approach is better explored on larger models. For example, for MoCo (He et al., 2020), the performance gain with ViT-B on Src. and Tar. are 21.03 and 11.98 respectively, while they are 1.95 and 2.86 with RN50. For Dino, from the Linear Clf., the model with larger patch size can facilitate generalization on target domains but has slightly weaker score on source domain. However, WIDIn can consistently improve the performance and minimize the gap between ViT-B$_{16}$ and ViT-B$_8$,

## C.3 LEARNING VISION ENCODER FROM SCRATCH

In addition to use image encoder which is trained on an image-only dataset or a multi-modal dataset as intialization, we can also train an image encoder from scratch for domain generalization. However, as the training data is very limited, training the image encoder from scratch is very easy to overfit to the train set and the generalization of test data from the same domain is still poor.

In detail, on CUB-Painting, we train a ResNet18 from scratch via fully-supervised learning, the test accuracy (Src. 22.45, Tar. 14.59). Then, we test our approach after performing multi-modal contrastive learning between language embedding extracted by CLIP-Text and the image embedding to update the image encoder from scratch, and get the results (Src. 24.85, Tar. 15.30). Though it is still less acceptable, we can still see a slight gain by our approach.

## D   EXPERIMENT IMPLEMENTATION

**Availability of Datasets**. All of the datasets used for experimental study in our approach are publicly available.

### D.1   TRAINING DETAILS

As mentioned before, our approach can be applied to the joint vision-language embedding space can be learned via either [a] large-scale multi-modal pretraining such as CLIP or [b] finetuning from two uni-modal models on the existing training data $\mathcal{D}_s$ where the uni-modal model indicates the model which is only trained on dataset of one modality.

Thus, for fair comparison in [a], all approaches reported in Table 1 uses ViT-L$_{14}$ for experiments. However, for [b], as finetuning ViT-L is implausible given limited computational resources, we thus choose ViT-B$_{16}$ for validation. (ViT stands for Vision Transformer (Dosovitskiy et al., 2020) where "L" and "B", indicating the size of ViT, stand for large and base respectively. For ViT, each image is first split into non-overlapping grids and each grid is called a patch. Then, the number at sub-index means the size (*i.e.*, height and width) of patch.)

During network training, we experimentally found that using SGD optimizer to minimize the alignment constraints based on contrastive learning is beneficial for the final performance. As such, even though the CLIP model is pre-trained by using AdamW optimizer, when we train the projector $\mathbf{F}_P$ by minimizing the loss $\mathcal{L}_{ia} + \mathcal{L}_{ca}$, we still use SGD as the optimizer with 0.002 as the learning rate. Then, when we minimize the feature prediction loss $\mathcal{L}_{feat}$ and classification loss $\mathcal{L}_{cls}$, we use the AdamW as the optimizer with the learning rate 0.0001.

To balance the network optimization strength from different losses, we set the weight of $\mathcal{L}_{feat}$ as 2.0 while keep the weights for all other losses as 1.0. The training is very efficient and the maximum number of epochs we used for training is 60. Meanwhile, from Eq. 3, $\mathbf{x}_e = \mathbf{x} - (\mathbf{t}_x - \mathbf{t}_c) = \mathbf{x}_e = \mathbf{x} + (\mathbf{t}_c - \mathbf{t}_x)$. Then following the definition of residual learning where $\mathbf{x}_e = \mathbf{x} + \mathbf{F}_D(\mathbf{x})$, the output of disentangler $\mathbf{F}_D(\mathbf{x})$ is trained to mimic $\mathbf{t}_c - \mathbf{t}_x$.

Besides, as in [b] for finetuning two pre-trained uni-modal models under contrastive learning, we found using AdamW optimizer is more stable and we set the learning rate as 0.0001. Meanwhile, since a larger batch size is needed to ensure the efficiency of contrastive learning, when finetuning the two uni-modal models before applying our approach WIDIn, we train the network with a batch size of 512 for 90 epoches on four V100 GPUs. After the finetuning is done, we then apply our WIDIn for domain generalization.

### D.2   PERFORMANCE DETAILS

**DomainNetMini** consists of one source domain and three targets domains during evaluation. In Table 1, we report the average of performance on three domains under Tar. and we below details the results on all three domains for reference (95.99, 95.81, 96.21).

Meanwhile, for our approach, the groundtruth of feature disentangler output is still in the same vision-language embedding space. In other words, both ours and Linear Clf. learns a linear classifier on the joint vision-language embedding space, while the only difference is that our approach first introduce a feature disentangler to predict the domain-invariant representation from original visual embedding but still reside in the same embedding space. Thus, the performance of Linear Clf. naturally serves as an ablation study for the embedding type.

Then, the baseline MLP Clf. shares the same architecture as our WIDIn where both $F_D$ and $F_C$ are learn for image classification. However, no loss is applied on the output of $F_D$ and the entire network is only trained by minimizing the cross-entropy loss.

Then, for ablation study on three embedding types in Table 3, we note the discriminative information for classification may also be useful in indicating the domains, for example, the eyes of bird is natural images is smaller than that in painting/drawing. As such, the domain-specific discriminative information in domain-invariant representation can also be useful for domain recognition. However, the experiments for object class recognition can still clearly demonstrate the effect of our approach.

### D.3 MORE EVALUATION ON CLIP EMBEDDING SPACE

Besides the experimental validation on CUB-Painting and DomainNetMini where domain bias are defined by the class-agnostic environment (*e.g.*, both dogs and cats may appear in the natural environment and part), we also evaluate our approach on Waterbirds (Sagawa et al., 2020), the domain bias is defined by the connection between class and environment.

**Waterbirds** (Sagawa et al., 2020) consists of two classes, *i.e.*, waterbirds and landbirds, but under different environments. During training, the environmental context is always matched with the class of birds (source), *i.e.*, the landbirds are always on the land and the waterbirds are always on the water. However, during testing, there are images with mismatched environments (target), *i.e.*, landbirds on the water and waterbirds on the land. For evaluation purpose, we calculate the accuracy for each class at each environment. In this way, a domain is essentially determined by the class name and the environment. In other words, the model has only seen two domains at train time but four domains at test time. Ad such, we first detail the performance of zero-shot baseline and our approach below.

Table 9: Detailed results on WaterBirds.

| Approach | Zero-Shot | WIDIn(Ours) |
|---|---|---|
| landbirds on land | 99.45 | 99.31 |
| waterbirds on water | 97.64 | 98.70 |
| landbirds on water | 48.87 | 52.84 |
| waterbirds on land | 66.51 | 92.81 |

Table 10: Comparison with Baseline.

| Approach | Src. | Tar. |
|---|---|---|
| Zero-Shot | 98.55 | 57.69 |
| Zero-Shot* | 98.55 | 58.77 |
| Linear Clf. | 92.75 | 72.80 |
| WIDIn | **99.00** | **72.83** |

In Table 10, the approach "Zero-Shot*" means adding domain description when extracting the language embedding for zero-shot classification. For example, in Table 1, on DomainNetMini, we use "the photo/image of an airplane" for the approach "Zero-Shot" but use "the clipart of an airplane" for the approach "Zero-Shot*". On Waterbirds, the domain is determined by the class and the environment. Since the classification task is still to differentiate waterbirds from landbirds, we choose to fuse the language embeddings of descriptions of the same class, *e.g.*, averaging the language embeddings extracted from "The photo of a landbird in the forest" and "The photo of a landbird on the water" as the center of class "landbird". However, from Table 10, the performance gain is not significant.

Further, as the class names are defined by the environment, we think the CLIP language embedding, e.g., extracted from "The image of a landbird", may naturally include the environment information of land (water) to the class name landbird (waterbird). As such, the effect of domain-invariant visual embedding by WIDIn is not significantly shown comparing with visual embedding. Thus, we think our WIDIn can still be further improved by using other description to represent these two classes, e.g., the description of attributes of the objects such as size and shape, which is not explicitly defined by the environment but can still be used to distinguish these two bird classes. Then, our framework can still be naturally generalized to these. We would like to leave it for future work.

### D.4 EXPERIMENTAL DETAILS OF LONG-TAIL CLASSIFICATION

**Dataset.** 1)**ImageNet-LT** (Liu et al., 2019) is a long-tailed subset of the ImageNet (Deng et al., 2009). There are in total 115.8K images from 1000 categories and the number of training instances ranges from 5 to 1280. 2) **iNaturalist** (iNatrualist, 2018) contains around 437.5K images from 8142 categories and the number of training instances per class ranges from 2 to 1000. For both ImageNet-LT and iNaturalist datasets, the distribution of test sets is uniform where the numbers of test data per class are 50 and 3 respectively.

**Evaluation Metric.** We directly report the performance on all classes. Meanwhile, according to the size of train set, we split the classes into three groups, Many ($\geq 100$ samples), Medium (Med, $20\sim100$ samples) and Few ($\leq 20$ samples). We break up the accuracy over the groups and the standard deviation of accuracy over all groups (STD) are reported.

In long-tail classification, a model may show bias towards the major classes with sufficient number of training data and thus ignore the minor class with limited number of training data. By removing the domain-specific information from the features, we expect the model could be robust to the bias caused by the data distribution. Meanwhile, as we fix the CLIP vision encoder during the entire

Table 11: Comparison on long-tail learning.

| Approach | Few | Med | Many | STD |
|---|---|---|---|---|
| Zero-Shot | 63.63 | 66.43 | 66.86 | 1.75 |
| VL-LTR (Tian et al., 2022) | 59.30 ($\downarrow$ 4.33) | **74.6** | **84.5** | 12.70 |
| Ours | **68.91** ($\uparrow$ **5.28**) | 71.69 | 72.07 | **1.72** |

training process, as mentioned before, we follow LogitAdj (Menon et al., 2020) to add class-based margins to the loss during training ans also add one linear layer with the same input-output dimension to project & adjust features. The class margins are calculated based on the data distribution and first theoretically studied in LogitAdj (Menon et al., 2020).

Meanwhile, as listed in Table 11, comparing with VL-LTR (Tian et al., 2022) which update the backbone and use extra dataset for training, we fix the vision encoder after being initilized with the CLIP-pretrained model and do not need access to any extra instance-specific description, instead, we just build description for each class. Additionally, VL-LTR (Tian et al., 2022) achieves significant performance on the Many group at the expense of sacrificing accuracy on Few group, comparing with zero-shot baseline. In contrast, our approach is capable of achieving consistent performance gain across all class groups and also reducing the accuracy variance across these three groups.

**iNaturalist** is released for a long-tail learning challenge every year starting from the year of 2017. We follow the most common evaluation setup and use the iNaturalist dataset released on 2018 for the experiments. Due to the limitation of table resize, we report the detailed performance at each class group here.

Table M1: Detailed results on iNaturalist.

| Approach | Backbone | Few | Medium | Many | Overall |
|---|---|---|---|---|---|
| LWS (Kang et al., 2020) | RN50* | 65.50 | 66.30 | 65.00 | 65.90 |
| LogitAdj (Menon et al., 2020) | RN50* | 66.27 | 66.34 | 66.79 | 68.44 |
| TSC (Li et al., 2022) | RN50* | 67.80 | 70.60 | 72.60 | 69.70 |
| Zero-Shot (Radford et al., 2021) | RN50[†] | 2.91 | 3.43 | 3.48 | 3.41 |
| Baseline | RN50[†] | 69.90 | 71.81 | 73.51 | 72.60 |
| WIDIn(Ours) | RN50[†] | 71.68 | 73.03 | 73.56 | 73.20 |
| Zero-Shot (Radford et al., 2021) | ViT-B$_{16}^{†}$ | 3.66 | 4.35 | 4.16 | 4.16 |
| Baseline | ViT-B$_{16}^{†}$ | 72.11 | 73.77 | 73.97 | 73.72 |
| WIDIn(Ours) | ViT-B$_{16}^{†}$ | 75.01 | 75.00 | 75.75 | 75.45 |

*: Trained from scratch. [†]: *Fixed* and initialized with CLIP (Radford et al., 2021).

For the results by LogitAdj (Menon et al., 2020) on iNaturalist in Table 2 in the main paper, it is realized by additionally including adaptive loss, serving as an ablation study, but it is not the main technical contribution from LogitAdj. As such, we use the performance by only using the loss designed in LogitAdj for comparison.

### D.5 ABLATION STUDY ON TRAINING SCHEDULE

For our WIDIn, we first only learn the projector under the alignment constraint. Then, we learn the feature disentangler and classifier separately, *i.e.*, the embedding $\mathbf{x}_e'$ predicted from $\mathbf{F}_D$ is not used to train $\mathbf{F}_C$. We represent the schedule as ($\mathbf{F}_P$, $\mathbf{F}_D$, $\mathbf{F}_C$). We also provide ablation study on the training schedule in Sec. 6.1.

As shown in Table 13, in practice, we found that separating the training of the three modules result in the best performance. After all, at the early stage of the network training, the projector for predicting token embeddings has not been properly learned. Then, using the $\mathbf{x}_e$ at early stage to guide the training of predictor ($\mathbf{F}_P+\mathbf{F}_D$, $\mathbf{F}_C$) will confuse the network training and result in model collapsing. Similarly, if the classifier is jointly trained with the predictor ($\mathbf{F}_P+\mathbf{F}_C$, $\mathbf{F}_D$), the model may show more bias to the source domain. Meanwhile, if we use both $\mathbf{x}_e$ and $\mathbf{x}_e'$ to train the classifier ($\mathbf{F}_P$,

Table 13: Ablation on Training Schedule

| Strategy | Src. | Tar. | Avg. |
|---|---|---|---|
| $(\mathbf{F}_P+\mathbf{F}_C, \mathbf{F}_D)$ | 87.21 | 67.25 | 76.98 |
| $(\mathbf{F}_P+\mathbf{F}_D, \mathbf{F}_C)$ | 86.37 | 64.82 | 75.59 |
| $(\mathbf{F}_P, \mathbf{F}_D+\mathbf{F}_C)$ | 86.76 | 66.10 | 76.43 |

$\mathbf{F}_D+\mathbf{F}_C$), the performance may drop slightly. As such, we only use the $\mathbf{x}_e$ to train the classifier. In this way, as we are separating $\mathbf{x}_e$ and $\mathbf{x}'_e$ for loss calculation, we can learn the $F_C$ and $F_D$ at the same time. Meanwhile, when we train the $F_P$, we use the direction loss in LADS (Dunlap et al., 2022) as an alternative of MSE loss, and it underperforms our current design as Direction loss (CUB-Painting, Src.: 86.81 and Tar.:66.42) ignores the consistency of vector length.

