# OpenReview forum: "Wording Image for Domain-Invariant Representation in Domain Generalization"
_ICLR.cc/2024/Conference — ICLR 2024 Conference Withdrawn Submission_

### Official Review · Reviewer_wwhy · 2023-10-24

**Soundness:** 2 fair
**Presentation:** 3 good
**Contribution:** 2 fair
**Rating:** 5
**Confidence:** 3

**Summary:**

This paper studies domain generalization, where the key challenge is to learn domain-invariant visual representations for each category. The authors argue that language embeddings of a particular category are naturally domain-invariant. Moreover, the difference between the pseudo-language embedding (prompted with the input image) and the original language embedding (prompted with the class description) represents the domain-specific counterpart. To this end, the authors propose WIDIn to learn domain-invariant visual representations using language embeddings. Empirical evaluation under various domain generalization benchmarks demonstrate the effectiveness of the proposed method.

**Strengths:**

1. This paper is well-written and easy to follow.
2. The motivation is intuitive and seems reasonable.
3. The figures vividly illustrate the pipeline of the proposed method, especially the left part of Figure 1 and Figure 2b.
4. The improvements over baselines are relatively significant.
5. The proposed method is also capable of long-tailed image classification.

**Weaknesses:**

1. **Lack of evidence to support the core insight.**  As illustrated in Figure 2b, $t_x - t_c$ is parallel to $x - x_e$, but in Figure 3, no such phenomenon can be observed. It is encouraged to draw a parallelogram composed of $x, x_e, t_x, t_c$ for each category both on the source domain and the target domain.

2. **Missing ablations.** There are three objectives in total, including $L_{ia}$, $L_{ca}$, and $L_{feat}$. It is better to study the effectiveness of each component step by step.

3. **The underlying motivation of $L_{ca}$.** Specifically, $L_{ca}$ aims to minimize the distance between the domain-specific text embedding and the domain-invariant text embedding. However, as the optimization goes on, the difference between these two embeddings becomes small, which is used to measure the domain-specific parts. However, domain-specific parts always exist and *never* go small. Therefore, minimizing $L_{ca}$ becomes strange to me.

**Questions:**

I do not have further questions, please refer to the weaknesses section.

---

### Official Review · Reviewer_m3y6 · 2023-10-31

**Soundness:** 3 good
**Presentation:** 2 fair
**Contribution:** 3 good
**Rating:** 6
**Confidence:** 3

**Summary:**

The paper introduces a method called WIDIn, which connects visual and language information to create domain-invariant features. This is achieved by representing images as word tokens and using the difference between the image's and its class description's language embeddings to promote domain-invariant representation learning. Experimental results show the effectiveness of WIDIn on benchmark datasets.

**Strengths:**

1. The proposed method is novel. With the added F_p, F_D, and F_C, it add learnable parameters to LADS and thus lead to better performance.
2. The experimental results show the power of proposed method.
3. Ablation study with different prompts, contrastive learning w/ w/ot labels, and training/freezing language model is interesting.
4. The extension experiments on long tail case also better support the power of the proposed method.

**Weaknesses:**

1. Is the image encoder trainable? If so, it might be unfair to compare with Linear Clf. and MLP Clf.
2. It would be great if the author could compare to other more general methods in domain generalization, where only the training domain is available and the domain descriptions of testing domains remain unknown.
3. It would be great if the proposed method could be evaluated on bigger benchmarks (DomainNet and Office-Home) with more domains as the proposed methods do not require any access to the target domain.
4. The presentation of this paper is a little bit awkward. For example, the training details, such as loss for each step, are not presented in the main paper.

**Questions:**

See weakness

---

### Official Review · Reviewer_z1uR · 2023-11-01

**Soundness:** 2 fair
**Presentation:** 3 good
**Contribution:** 2 fair
**Rating:** 5
**Confidence:** 5

**Summary:**

Authors propose to project images into language space by representing each image as a word token, which is attached with hand-crafted prompt and fed into language encoder, where the difference between the extracted embedding and the language embedding of its class description is used to estimate the domain-specific counterpart, which facilitates the domain-invariant representation learning. Experiments demonstrate the effectiveness of this approach.

**Strengths:**

1. Reported results outperform baselines: experimental studies on two domain generalization benchmark datasets and two long-tail benchmark datasets demonstrate the effectiveness of this approach.
2. The presentation is clear and easy to follow.

**Weaknesses:**

1.  The novelty is limited. Representing an image as a word token is very common in many recent multimodal models (e.g., BLIP, LLava, CM3, RA-CM3, CM3Leon, etc.). The overall idea and pipeline is still similar to LADS. The difference is that, this paper learns from domain-invariant representations while LADS learns from domain-specific representations, and the way a domain-specific or domain-invariant is obtained is similar.

technical details:

2. From my understanding, $t_x$ is the "unified" representation of domain-invariant and domain-specific features. In this case, I wonder why you use a text encoder to encode "an image of [V]" to represent $t_x$? Given that you claim "an image" represents an invariant domain, it would involve some biases (towards invariant domain) this way. Why not directly encode the [V] token alone?

3. I am concerned about using "image" to represent the domian-invariant space. Have you tried using other words to replace "image"? For example, using "painting" instead: projecting everything into the painting-domain. I doubt "an image of {...}" works due to the authors' claim that, "image" corresponds to an invariant space. I guess other words could have the similar results.

4. Code is not provided.

**Questions:**

please see weaknesses